# Using Qualitative Methods to Explore Farrier-Related Barriers to Successful Farriery Interventions for Equine Welfare in India

**DOI:** 10.3390/ani9050252

**Published:** 2019-05-18

**Authors:** Dinesh S Mohite, Chand S Sheikh, Saurabh Singh, Jogen Kalita, Shereene Williams, Polly C Compston

**Affiliations:** 1Functional and Comparative Genomics Department, University of Liverpool, Liverpool L69 7ZB, UK; 2Haffkin Biopharmaceutical Corporation, Pune 411018, M.S., India; chand25b@gmail.com; 3Brooke India, A 223-226, Pacific Business Park, Sahibabad Industrial area, Ghaziabad 201010, U.P., India; saurabh@thebrookeindia.org; 4KABIL Office, Flat No.3 C, BD Enclave, Khanamukh, Guwahati-781014, A.S., India; jogenkalita@yahoo.co.uk; 5Brooke, 5th floor, Friars Bridge Court, 41-45 Blackfriars Road, London SE1 8NZ, UK; shereene.williams@thebrooke.org; 6Royal Veterinary College, Hawkshead Lane, Hatfield, Hertfordshire AL9 7TA, UK; pcompston@rvc.ac.uk

**Keywords:** farriery, working equids, hoof, socioeconomics, qualitative

## Abstract

**Simple Summary:**

Farriery is important for maintaining equine (horse, mule and donkey) foot health, but is often poorly-executed in low and middle-income countries like India. It is important to understand the challenges in providing farriery services if external agencies want to improve them. This study, undertaken in North India, started to build this understanding by collecting information from farriers’ points of view. Through focus group discussion, it became clear that farriery in this area was very demand-led: farriers found it difficult to practice improved techniques if animal owners were not prepared, or unable, to pay for the extra time that this takes. This was linked to the self-perceived status of farriers within the community: those with higher status (often due to having additional income streams) were more confident in their interactions with animal owners. Difficulties associated with demand were closely linked to the external environment: farriers whose work relied on the local brick kilns had been significantly affected by a recent down-turn in that industry. Training and technical knowledge varied; training was more popular when it was inclusive and addressed the questions farriers had about their work. Understanding this system from the perspective of the stakeholders within it is essential for successful improvement programmes.

**Abstract:**

Farriery is a critical component of healthcare services for working equids. However, in India, an informal workforce, lack of structured training facilities and non-implementation of farriery regulations pose challenges for quality farriery. Brooke India, an equine welfare organisation, has undertaken many initiatives aiming to improve farriery services, including technical training and engagement with equid-owning communities. However, this has met with varying success. The study aimed to identify factors that prevent farriers providing quality farriery services. Focus-group discussions were conducted with farriers from two districts of Uttar Pradesh with varying programme outcomes. Within each area, farriers were grouped according to previous level of engagement with Brooke programmes. Demand for services, farrier status, the external environment and technical training and knowledge were identified as key elements that affected farriery work. These factors were very context specific: in areas where brick kilns provided the majority of customers, recent closures had resulted in an increase in those farriers’ feeling of insecurity. A systems approach to improving farriery services, taking these factors into account, is advised. Mentoring-based capacity building, which is closely aligned to farrier needs and expectations, is expected to have positive results in terms of technical skill and farrier engagement.

## 1. Introduction

There are approximately 1.1 million equids in India [1]. As in other low and middle-income countries (LMICs), equid ownership is an important livelihood strategy for many families and therefore these animals’ health is paramount [2,3]. Lameness is one of the most common health problems observed in working equids, with a prevalence of up to 100% [4,5,6]. Independent of the initial aetiology of lameness, often first-line treatment involves rest and non-steroidal anti-inflammatory medicine. However, this medical management in working equids is often compromised because they must work hard every day without the opportunity for time off and have limited access to animal healthcare services. Therefore, lameness is a major source of pain and compromised welfare in working equid populations [4,5,6,7]. In equids, lameness is often caused by foot pain; this is where the proverb “no foot, no horse” originates from. Appropriate hoof care (farriery) is an essential part of maintaining equine soundness; inadequate farriery can cause lameness as well as exacerbating existing limb pathology [5].

Farriery as a profession is often underdeveloped in LMICs [7,8]. Historically, interventions and recommendations by non-governmental organisations to address poor farriery have often focused on providing free services imported from the West, or semi-intensive technical training programmes, potentially with some form of owner involvement [6,7,9]. However, little work has been published to investigate the root causes that exist preventing farriers from achieving a higher level of technical competence, and care must be taken to ensure that community involvement is inclusive and participatory [10]. Recently, the importance of understanding human behavior change in relation to equine welfare has been highlighted [11]. Qualitative methods of data analysis can provide a deeper understanding of local issues, allowing recognition of how animal owners define and prioritize problems that are affecting them. These methods include thematic analysis. Thematic analysis is a method used in qualitative research to analyse qualitative data and identify patterns or themes across a dataset that answer the research query in a meaningful way. These methods are useful to unfold complex issues and gain increased understanding of peoples’ perceptions, opinions and ideas, as well as where consensus and disagreement exists [12,13].

Since 1992, Brooke, an equine welfare organisation, has been working in India, where poor farriery practices are commonplace due to multifaceted, complex reasons [10,14]. There is a complicated set of factors determining the supply of, and demand for, farriery services. Animals are often shod in urban centres during a working day, creating time pressure on the farrier from the owner and therefore an environment where good-quality trimming is not possible. Profit margins for farriers are small, as they are for the equid owners, making opportunity costs high; this means that it is often not practical for farriers to spend an appropriate amount of time trimming/shoeing the feet of any one animal. Whereas structured professional systems exist in some countries that reflect the depth of technical knowledge required for high-quality farriery (e.g., the Worshipful Company of Farriers in the U.K.), in LMIC contexts, farriery is often performed by unskilled informal workers with little or no training, in a trade that is often handed down from father to son, and that lacks professional regulation [15]. There is little understanding of the hoof’s anatomy and limited access to appropriate tools. Owners need to be able to recognise the importance of hoof balance and good quality farriery in order to ask for it [16]. Ultimately, equid owners must be willing and able to pay more for a service that takes more time.

Brooke India has undertaken many initiatives aiming to address these issues and improve farriery services, including technical training and community engagement [17]. Training includes demonstration of good farriery, practical teaching sessions and knowledge exchange meetings. Community engagement has included linking local farriers with existing equine welfare groups within communities that Brooke India works with and discussing hoof care with equid owners. However, these interventions have met with varying success. Hoof shape parameters (as measured by Brooke’s Standardised Equine-Based Welfare Assessment [18]) have not improved. This qualitative study aimed to explore the reasons for the varying outcomes following Brooke India farriery interventions and understand the factors underlying farrier motivation.

## 2. Materials and Methods

### 2.1. Ethical Review and Pretesting

The research was approved by Brooke’s Animal Welfare and Ethical Review Body. The discussion guide was pretested and assessed for time, question suitability and completeness. Information collected during pre-tests was excluded from final analyses.

### 2.2. Study Area

Data were collected in September 2016 in two districts of Uttar Pradesh, India: Meerut and Muzaffarnagar (Figure 1). These are densely populated areas, known for agricultural production, especially sugarcane and wheat. Equids are mainly employed to transport raw bricks for the brick-kiln industry and to carry goods and people to and from market places. Muzaffarnagar and Meerut were purposefully selected for this qualitative cross-sectional study. Brooke India has been working in both for a similar period, and they have a broadly similar context (as described above). However, Muzaffarnagar has had no improvements in hoof shape following Brooke intervention, whereas Meerut has had the most positive results of Brooke India’s intervention areas.

### 2.3. Farrier Selection

The farriers in each area were categorised into two groups by unit staff based on subjective experience:

“Engaged” farriers, who were those who willingly participated in Brooke India’s engagement events, seemed ready to adopt improved practices and were linked with community-based organisations/institutes.

“Less-engaged” farriers who did not seem to want to engage with Brooke India’s events or want to change their practices and were not linked with community-based organisations.

The farriers’ locations were mapped, and areas identified to allow the farriers to access, and participate in, group discussions easily. Exclusion criteria included any farriers who did not complete the informed consent process, participants under 18 years of age and any situation where carrying out group discussion meant that animals and owner are kept waiting for farriery services.

### 2.4. Focus Group Discussion Guide

A focus group discussion guide (Table 1) was developed that explored two main research questions:What are the most significant limiting factors that prevent farriers putting increased technical knowledge into practice?What are the differences in terms of challenges of farriery and engagement with Brooke’s programme in these two different locations?

### 2.5. Data Collection

Data were collected in September 2016. Each focus group discussion was prearranged with participants to ensure appropriate timing and location. The lead researcher (DM) conducted the group discussions. Oral informed consent was obtained from all participants (Appendix A). Discussions were conducted in Hindi and recorded, then transcribed and simultaneously translated by the lead researcher (DM).

### 2.6. Data Analysis

Data were entered and saved in Microsoft Word and backed up on Brooke’s server and an external hard disk. Data were coded and key themes were identified. During the analysis, it became clear that these themes were cross-cutting across all four groups and therefore as they developed, data from the four groups were consolidated. These themes were reviewed iteratively, initially by DM and PC, and then discussed with co-authors to ensure that the resulting conceptual framework was embedded within contextual knowledge.

## 3. Results

Four focus group discussions were carried out (one for each category of farriers in each district). Each discussion included between six and nine participants. In total thirty-two farriers participated. These four groups were categorized as A (Meerut; engaged group), B (Meerut; less-engaged group), C (Muzaffarnagar; engaged group) and D (Muzaffarnagar; less-engaged group). A conceptual framework was developed by integrating results from all four discussion groups (Figure 2). The elements of this framework are discussed below.

### 3.1. Demand for Services

The way in which farriers interacted with equid owners played an important role in the way that they conducted their business. Farriery was described as being very demand-driven, with farriers reporting that they were often being asked directly by owners to perform poor quality work (especially frog removal and ‘dumping’, where the toe is removed).
“Sometimes the owner tells us to trim the whole frog, otherwise he will go to another farrier.”*—Farrier No. 2, Group C*

Owners were not perceived to want or expect quality shoeing due to the length of time it would take, and farriers did not feel able to initiate these discussions. This was reinforced because owners preferred farriers who were quicker. Additionally, owners were unwilling to pay any extra for an improved service.
“70 to 90% equid owners ask us to do farriery while the animal is still harnessed to the cart.”*—Farrier No. 8, Group C*
“Clients do not pay us well. Even if we use a specialised shoe to correct a hoof-related issue, they will not pay any more.”*—Farrier No. 5, Group B*

The lack of demand for improved services was exacerbated for older farriers, who had built up individual relationships with owners who expected the service, and price, that they had always received from them.
“I am the renowned and oldest farrier in our area. If I did poor farriery in that case the client will come [back] to me only because they trust me.”*—Farrier No. 7, Group A*

It became clear that these issues around demand were particularly acute in Muzaffarnagar (see Section 3.3: The External Environment). Both groups of farriers who were less-engaged with Brooke’s work described competition between farriers as a problem, leading them to accept any business that they could, and even offer their services on credit. Farriers would often undercut each other’s prices. An additional problem described in Muzaffarnagar was that the equine welfare community groups formed by Brooke India had been able to exert commercial power over the farriers.
“Clients are our boss because they formed equine welfare groups and they have received more knowledge than us.”*—Farrier No. 3, Group D*

### 3.2. Farrier Status

Owner demand was linked to the farriers’ status in the community. Many of the farriers saw no social recognition for their profession. This was exacerbated in Muzaffarnagar.
“[There is] no respect in this job, people say we hold the horse foot, which they perceive as bad.”*—Farrier No. 5, Group D*

Despite this, some held the belief that they were among the best farriers in their area. This resulted in a reluctance to adopt improved farriery practices.
“I already doing good farriery and do not need to learn.”*—Farrier No. 7, Group A*

Younger farriers were more receptive and readier to adopt new farriery practices.
“I want to learn more, I am not satisfied with what I have learnt.”*—Farrier No. 3, Group B*
“I want information regarding brushing. Also, I want to learn about hoof balancing.”*—Farrier No. 6, Group B*
“Earlier we only need to think about our own profit, now we also think about welfare of equines.”*—Farrier No. 2, Group B*

A noticeable difference, which mainly applied to the farriers in Meerut, applied to farriers that had more than one income stream. These farriers tended to be more confident in their ability to provide a superior service as they had alternative resources available.
“I never agree with owner, if the client wants to go and avail another’s services, then I don’t bother as I have other business.”*—Farrier No. 4, Group B*

Most of these farriers were employed in equid-related businesses (working in brick-kilns, equine hair clipping/saddlery/saddle shops and ceremonial use of equids); others had motor vehicles that they used as taxis. Several farriers who also made shoes, and the use of good quality shoes was reported as an advantage for attracting clients.
“We use good quality shoe [heavy metal] to attract clients.”*—Farrier No. 3, Group C*

### 3.3. The External Environment

Many farriers reported that farriery as a business was becoming more difficult. Inflation had caused an increase in the cost of shoes and tools. Farriers also communicated that they were unable to obtain a loan from banks to support their business. Much of this was attributed to changes in the brick kiln industry, where many of their clients were working. The seasonality of brick kiln work (November to June) was a continuing problem for farriers who were reliant on custom from brick kiln workers. However, the period that the kilns had been open during the season prior to data collection had been four instead of eight months due to the government enforcing air pollution law and banning kiln operation.
“In last 3 years, the farriery work has reduced to 75%.”*—Farrier No. 4, Group C*

The farriers in Muzaffarnagar were particularly affected by this decrease in equine industry, which had exacerbated the difficulties associated with demand-driven services and high levels of competition in the area.
“We will die hungry, if this persists.”*—Farrier No.8, Group C*

### 3.4. Training and Technical Knowledge

There were three main pathways to becoming a farrier. Some had learnt it as a trade from a local ‘ustad’ (community teacher). Others had learnt from family members or were self-taught. The more engaged farriers tended to have been taught by ustads, whereas a characteristic of family or self-taught farriers was that they had not spent very long learning their trade. There was also a split between the two different locations; farriers from Muzaffarnagar were more likely to be self or family-taught.
“I am working in brick kiln as a beldar [construction worker]. I do not have much to do after I finish my work in the afternoon, so I thought I could shoe the equids that work at the brick kiln. Thus, I learnt farriery in one and a half to two months.”*—Farrier No.1, Group D*

The majority of the farriers expressed an interest in developing their skill in farriery. The most common challenges that they described were particular pathological presentations (Table 2). Many were interested in understanding the cause and treatment of overgrown hooves, bruised sole, overreaching, forging, interference or brushing and hoof balance. In the brick kiln off-season, the owners did not use farriers, resulting in overgrown hooves that were difficult to correct over the course of the working season.
“It takes more time to trim hooves of equine as they overgrow in brick kiln off season.”*—Farrier No. 2, Group A*

An additional problem that they reported facing was working with “aggressive” animals because it is difficult to handle them, which results in needing more time to trim and shoe them. Therefore, they wanted to learn more about animal handling.

The farriers reported mixed feedback on their interactions with Brooke India. As expected, those with the most positive feedback were the engaged farriers. They had experienced a range of activities facilitated by Brooke India, for example, *in situ* training, classroom sessions, exposure visits to other farriers, competitions, participating in equine welfare group community meetings and attending equine fairs. However, even the most engaged farriers, along with farriers from other groups, did not feel as though the Brooke India programme always addressed the challenges that they were facing. They felt that they knew the basic trimming techniques that were being taught and would find training on how to address the specific pathologies mentioned earlier more useful.
“Brooke Farrier asks questions from us but never tells us anything or gives any information.”*—Farrier No.7, Group A*

## 4. Discussion

These results show that farriers working with working equids in Uttar Pradesh are likely to have competing priorities that affect their ability to improve their technical skill. Using the COM-B (capability, opportunity, motivation and behaviour) model of behaviour change [19], this seems to be due to a mixture of inadequate opportunity; equid owners being unable to pay for, or unwilling to allow, improved farriery to be performed on their animals; and a lack of motivation associated with their socioeconomic status and training. The importance of tailoring technical training programmes to participants’ wants and needs is highlighted.

The factors that influence a quality farriery service are highly context-specific, as evidenced by the clear difference seen between the two areas studied, despite their geographical proximity. Farriers in Muzaffarnagar were very dependent on trade from the brick kilns. In India, especially in Uttar Pradesh, the brick kiln industry employs many working equids to transport raw bricks. A recent government decision to ban the brick kilns because of their polluting effects [20] has had a significant effect on the livelihood of these equid-owning communities. The associated uncertainty influences the business of local farriers. The farriers who depend on brick kiln clientele are frustrated and demotivated. Many are worried about their future. Due to fewer clientele, there is more competition amongst farriers. It is not hard to understand why these farriers may find it more difficult to engage with external organisations and training programmes.

Farriery is a highly skilled profession that needs intensive training. However, this was not reflected in the interactions that the farriers described with their clients. Often owners will trust an old and renowned farrier, regardless of their technical ability, and conversely there is a view among farriers that equid owners do not understand good farriery. Animals are often shod on credit, and the demand for a quick, rather than quality, service is high. In many instances, owners refuse to remove the animal from the cart while presenting animal for shoeing, risking injury to the farrier, animal and owner. In general, the farriers do not feel that the farriery profession is respected or recognised within their community. In all groups, the relationship between the owner and farrier played an important role in the farriers’ practice. This emphasises the importance of involving the wider community, not just in generic training sessions, but in designing interventions that aim to improve farriery practice. Currently, there is no formal farriery professional body within India; curating opportunities for farriers to represent themselves as a professional body may improve the relationship that they have with their clients and promote a sense of their own expertise.

It is interesting that farriers with diverse income streams appeared to have increased confidence in their work and in their relationship with their clients, viewing their work as artisanal and refusing to provide what they saw as an inferior service. Diversification is a strategy that people worldwide often use to strengthen their livelihoods [21]. The ability to diversify can be influenced by people’s pre-existing socioeconomic status [22]. Many pro-poor policy interventions have aimed to support diversification; it is important that these consider social factors and are designed to target the appropriate groups [23,24]. In Lesotho, access to capital and business literacy were reported to be limiting factors for farriers to put skills gained during a training programme into practice [25].

Over the last 10–15 years, animal welfare organisations working in LMICs have put increased emphasis on community participation and needs assessment in order to achieve sustainable human behaviour change [12,26,27]. There are consequently fewer stand-alone training sessions, whether aimed at owners or people providing them with services (e.g., farriers, harness makers), recognising that these often do not lead to the changes intended [28]. This mirrors changes in the development sector, where agencies are increasingly taking a systems approach to address complex problems [29]. This work represents the first step towards developing an evidence base for future systems-level interventions. The integral role that brick kiln operation has for the farriers demonstrates this. Environmental, human rights and animal welfare concerns prevent recommendations that advocate for the brick kilns to remain open [30]. However, this is a continuing constraint for farriers that work to serve animals within the brick kiln industry. The difficult truth is that some of these farriers are experiencing negative market forces that will prevent them from being able to provide quality services. Animal welfare organisations must take care to map out the systems that they are working in order to articulate and implement a path to their ultimate goal [25]. For these farriers, future system mapping could include: data collection from animal owners to understand their relationship with farriery from their point of view, additional research with farriers to build on the understanding of their socioeconomic status, and work to understand if farriers who are more engaged and are in fact more technically competent. This work has limitations: it is highly context-specific since it only involved farriers who are currently working with Brooke India and therefore there may have been response bias introduced as Brooke India staff performed data collection. These are common problems associated with research in low-resource settings. Using animal-based and farrier competency assessment to triangulate results, and ensuring that results are fed back iteratively to the communities that they were collected from, will support the conclusions formed.

Where technical training remains part of interventions designed to improve farriery services, care should be taken to select participants who are engaged and therefore likely to implement new knowledge, and who the market will support. Resources are always finite, so they should be directed where they will make the most impact. Knowledge transfer programmes must be seen to be relevant for the participants in order to have a chance of being successful. Farriery is a complex subject; those responsible for delivering training must themselves be highly trained, as well as highly proficient in adult-training techniques for an audience with limited previous education [31]. This can lead to difficulties recruiting and retaining the right people for a training programme. Student-centred training methods have previously been used successfully in farriery, even extending the approach to include joint trainings and meetings with equid owners and local farriers [14,32]. Any capacity building will need continued mentoring where an interpersonal relationship needs to be developed between trainer and farrier for this to be effective. Brooke has developed mentoring frameworks to support its work with all animal health practitioners, including a specific framework for farriers with the aim of strengthening the entire animal healthcare system [33,34]. This framework outlines essential competencies and is used by Brooke staff to provide case-based mentoring. It is hoped that quality of learning is improved over didactic workshop sessions and issues beyond individual competency can be addressed. 

## 5. Conclusions

This study demonstrates the importance of listening to the challenges faced by the participants within a system of planning programmes to improve it. Farriery is a complex technical skill that exists within a multifaceted system: improvements to both must be addressed for sustainable changes to be made. A mentoring approach to technical capacity building is more likely to be successful than stand-alone training that is not tailored to the participants’ self-perceived learning needs. Farriery systems are context-specific and this must be taken into consideration during intervention design. Farriery has the potential to make a large impact on the welfare of working equids worldwide and should be an integral part of equine welfare initiatives.

## Figures and Tables

**Figure 1 animals-09-00252-f001:**
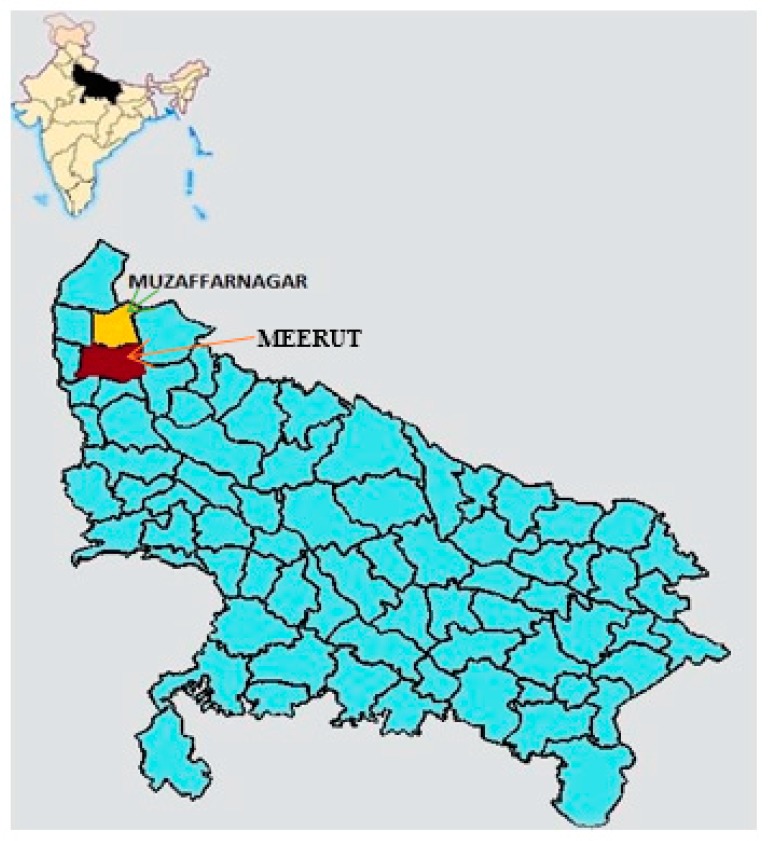
A map of Uttar Pradesh State indicating study sites.

**Figure 2 animals-09-00252-f002:**
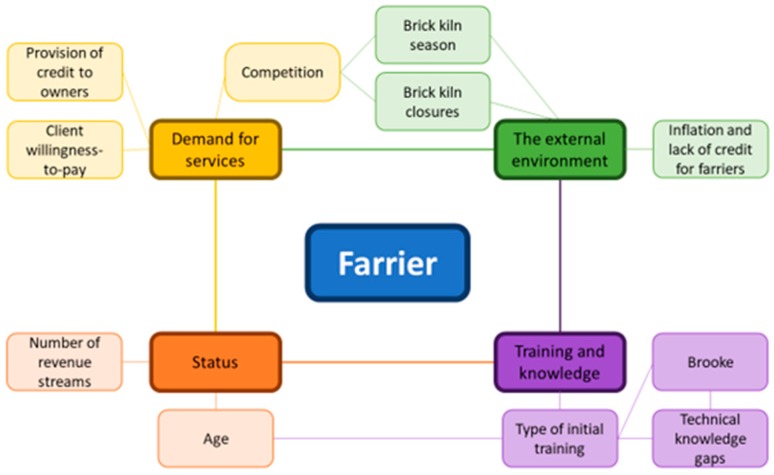
A conceptual framework depicting the key themes that farriers used to describe the factors influencing their work.

**Table 1 animals-09-00252-t001:** Focus group discussion guide.

Question Number	Questions
1	How long have you been practicing farriery?
2	How did you learn the farriery skill? *(examples-forefather/brother/local ustad/Brooke/other–specify, time took to acquire skill)*
3	Have you worked with Brooke? Explore in which way and for how long? *(Examples: Mentoring/training themes/exposures/association or Equine Welfare Group meetings, linking with community)*
4	How useful have you found working with Brooke? *(Explore when and how any interaction has been useful, as well as has not been useful in terms of cliental, economic, practices, social recognition, etc. Did they fulfil your expectations, what you wanted to learn?)*
5	Have you changed your practices after working with Brooke? *(Explore and use photographs: what are those practices, why or why not did change occur, how easy or difficult was it to change)*
6	What is the most difficult thing about your job? *(Explore: time, money, resources, competition, market, inflation, owner’s attitude pairwise ranking. For each, discuss how this will effect their job if it occurs more or less)* Top three difficulties recorded through PRA (participatory rural appraisal) exercise using pairwise ranking: ____________ ____________ ____________
7	Do you have any income streams apart from shoeing and trimming equine hooves? What are they? Include making shoes and tools. Which is the preferred activity? What would be your ideal job in 10 years’ time?
8	What is the best thing about your job and why? *(Explore: motivation behind this, any examples)*
9	How do you try to attract more clientele to your services? *(Explore: quick, cheaper, quality services, owners’ satisfaction)*

**Table 2 animals-09-00252-t002:** Top three challenges faced by local farriers (results of pairwise ranking).

Meerut		Muzaffarnagar	
(A) Engaged group	(B) Less-engaged group	(C) Engaged group	(D) Less-engaged group
Bruised sole (in equids)	Over-reaching (Dund marna)	Uncertainty/non-functional brick kiln industry	Fewer clients due to reduction in equine population
Handling aggressive equids	Interference or brushing	Unavailable or expensive farriery tools	Elderly farriers
Over-grown hooves in equids (especially in the brick kiln off-season).	Hoof balance	Bruised sole (in equids)	Low price paid for services

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
