# Peer review of "Using Qualitative Methods to Explore Farrier-Related Barriers to Successful Farriery Interventions for Equine Welfare in India"

_animals, 2019, doi:10.3390/ani9050252_

Round 1

Reviewer 1 Report

The work is an original idea  

However, I have some comments:

In general I miss some overall information: expecially Material methods shoul be rewritten and reported in a scientific way. This section and as consequence the results part, should be clearly rewritten. It lacks of a clear description on how the questions were conducted and the statistic that was performed. I suggest to the authors to divide the M&M section should be divided in Data collection ,Question formulation, Statistical analysis

I feel that the intention of this paper is sound but it need to be rewritten to be accted by the journal

Author Response

Dear Reviewer one

We would like to thank you for your suggestions and review that helps to revise the manuscript. Attached herewith point-by-point response for further action at your end.

Reviewer 2 Report

This study aimed to identify the reasons for the varying outcomes following Brooke India farriery interventions and understand the factors underlying farrier motivation. This is a very interesting paper, but needs little improvement.

In particular, I would add theFGD questionnaire in the main text to better clarify then  the results explained in the results section and connect the mat and met with the results. Moreover, I would cut the answers given by the ferries, in my opinion unnecessary.

Author Response

Dear Reviewer 2

We would like to thank you for review of the article that helped us to revise and improve the manuscript. The details of point-by-point response has been attached herewith in word file for your further action. 

Thanks & Regards.
